# PCK1 and DHODH drive colorectal cancer liver metastatic colonization and hypoxic growth by promoting nucleotide synthesis

Norihiro Yamaguchi[1†], Ethan M Weinberg[1†‡], Alexander Nguyen[1†§], Maria V Liberti[1], Hani Goodarzi[1#], Yelena Y Janjigian[2], Philip B Paty[3], Leonard B Saltz[2], T Peter Kingham[4], Jia Min Loo[1¶], Elisa de Stanchina[5], Sohail F Tavazoie[1,6]*

[1]Laboratory of Systems Cancer Biology, The Rockefeller University, New York, United States; [2]Gastrointestinal Oncology Service, Memorial Sloan-Kettering Cancer Center, New York, United States; [3]Colorectal Service, Memorial Sloan-Kettering Cancer Center, New York, United States; [4]Hepatopancreatobiliary Service, Memorial Sloan-Kettering Cancer Center, New York, United States; [5]Antitumor Assessment Core Facility, Memorial Sloan-Kettering Cancer Center, New York, United States; [6]Department of Medicine, Memorial Sloan-Kettering Cancer Center, New York, United States

*For correspondence:
stavazoie@mail.rockefeller.edu

[†]These authors contributed equally to this work

Present address: [‡]Perelman School of Medicine at the University of Pennsylvania, Philadelphia, United States; [§]Vatche & Tamar Manoukian Division of Digestive Diseases, David Geffen School of Medicine, University of California, Los Angeles, Los Angeles, United States; [#]Department of Biochemistry & Biophysics, Department of Urology, Helen Diller Family Comprehensive Cancer Center, University of California, San Francisco, San Francisco, United States; [¶]Cancer Therapeutics and Stratified Oncology, Genome Institute of Singapore, Singapore, Singapore

**Abstract** Colorectal cancer (CRC) is a major cause of human death. Mortality is primarily due to metastatic organ colonization, with the liver being the main organ affected. We modeled metastatic CRC (mCRC) liver colonization using patient-derived primary and metastatic tumor xenografts (PDX). Such PDX modeling predicted patient survival outcomes. In vivo selection of multiple PDXs for enhanced metastatic colonization capacity upregulated the gluconeogenic enzyme PCK1, which enhanced liver metastatic growth by driving pyrimidine nucleotide biosynthesis under hypoxia. Consistently, highly metastatic tumors upregulated multiple pyrimidine biosynthesis intermediary metabolites. Therapeutic inhibition of the pyrimidine biosynthetic enzyme DHODH with leflunomide substantially impaired CRC liver metastatic colonization and hypoxic growth. Our findings provide a potential mechanistic basis for the epidemiologic association of anti-gluconeogenic drugs with improved CRC metastasis outcomes, reveal the exploitation of a gluconeogenesis enzyme for pyrimidine biosynthesis under hypoxia, and implicate DHODH and PCK1 as metabolic therapeutic targets in CRC metastatic progression.

## Introduction

Colorectal cancer (CRC) is a leading global cause of cancer-related death. An estimated 145,000 Americans will be diagnosed with CRC and roughly 51,000 will die of it in 2019 (*National Cancer Institute, 2019*). The majority of these deaths are due to distant metastatic disease with the liver being the most common distal organ colonized (*National Cancer Institute, 2019*). While the prognosis of patients diagnosed with non-metastatic or locoregional disease is relatively better, a significant fraction of such patients will nonetheless experience subsequent metastatic progression. A key clinical need is identifying which patients will develop metastatic disease since few prognostic indicators of disease progression exist. Additionally, novel targeted therapies for the prevention and treatment of metastatic CRC are major needs.

**eLife digest** Colorectal cancer, also known as bowel cancer, is the second most deadly cancer in the United States, where it affects over 140,000 people each year. This cancer often spreads to the liver, in a process known as metastasis. To do this, the colorectal cancer cells must survive the low oxygen levels found in the blood that carries them from the gut to the liver, and in the liver itself. The majority of colorectal cancer cells that arrive in the liver die, but some survive leading to secondary tumors.

To investigate how the colorectal cancer cells that survive and metastasize to the liver accomplish this feat, Yamaguchi, Weinberg, Nguyen et al. took colorectal tumors from patients and introduced them into mice. This showed that tumors from patients with the worst outcomes tended to metastasize more efficiently in mice.

Next, Yamaguchi et al. looked to see which genes were active in the colorectal cancer cells that were able to metastasize and compared them to those that were active in the cells that could not. This analysis revealed that the gene coding for a protein called PCK1 was more active in the cells that could metastasize. In healthy cells, PCK1 promotes the generation of the sugar glucose, and Yamaguchi et al. observed that, in a low oxygen environment, higher levels of PCK1 allowed colorectal cancer cells to proliferate faster. Unexpectedly, this was due to PCK1 increasing the production of a molecule needed to make nucleotides, which are the building blocks for DNA and RNA. Consistent with this, metastasizing colorectal cancer cells generated more nucleotide precursors, and inhibiting the enzyme involved in nucleotide synthesis (DHODH) with an arthritis drug called leflunomide stopped colorectal cancer cells from spreading.

Metastasizing colorectal cancer cells depend on PCK1 and the nucleotide-synthesizing enzyme to grow, so therapies that target these proteins may help more patients to survive this kind of cancer. The findings also suggest that the arthritis drug leflunomide should be explored further as a potential drug for the treatment of colorectal cancer.

While cell lines established from human CRC have provided important insights into the biology of CRC, tissue culture drift, adaptation, and evolution can restrict their relationship to the pathophysiology of patient tumors (*Gillet et al., 2013*). Patient-derived xenograft (PDX) models, which allow for the growth of patient tumor samples in immunodeficient mice, can capture the endogenous diversity within a tumor as well as the patient-to-patient variability of metastatic cancer. Prior work has revealed that breast cancer, melanoma and non-small cell lung cancer PDX modeling can predict human clinical outcomes (*Quintana et al., 2012*; *John et al., 2011*). These models have revealed the potential of individualized mice models for prognostication and tailoring of therapies. Past CRC PDX models were subcutaneous implantations (*Cho et al., 2014*; *Gao et al., 2015*; *Oh et al., 2015*). Such subcutaneous PDX tumors are useful for tumor growth studies, but do not metastasize and are not exposed to the pathophysiologically relevant and restrictive conditions of the hepatic microenvironment. While orthotopic PDX implantation serves as a good model for studying metastasis in melanoma and breast cancer, this approach has limited feasibility in CRC, as the orthotopic tumor kills the host due to the obstructive dimensions that implanted tumors reach prior to liver metastatic colonization (*Quintana et al., 2012*). Thus, a clinically relevant PDX model of CRC that recapitulates the critical process of liver metastatic colonization is an important need.

Beyond predicting clinical outcomes and therapeutic responses, PDX models of mCRC could allow for identification of molecules that contribute to metastatic colonization through the use of in vivo selection. This process subjects a parental population (e.g. cancer cells, bacteria, yeast) to a severe physiologic bottleneck such that only those cells with the requisite gene expression states survive (*Fidler, 1973*). Such selection is repeated iteratively to enrich for cells that are best adapted to the new microenvironment. Molecular profiling can then reveal molecular alterations that enable the selected population to colonize the new environment (*Minn et al., 2005*; *Kang et al., 2003*; *Tavazoie et al., 2008*; *Pencheva et al., 2012*; *Loo et al., 2015*).

By engrafting patient-derived CRC primary and metastatic tumors of diverse mutational backgrounds, we observed that subcutaneous tumor engraftment efficiency and liver colonization capacity, but not subcutaneous tumor growth rate, was associated with patient overall survival. By

| Characteristics | Engraftment | No Engraftment |
|---|---|---|
| Subjects providing samples (n=31) | 15 | 16 |
| Total samples (n=40) | 17 | 23 |
| AJCC Stage at time of surgery | | |
| Stage II/III (n=8) | 5 | 3 |
| Stage IV (n=23) | 10 | 13 |
| Location of tissue sample | | |
| Colon (n=20) | 8 | 12 |
| Liver (n=16) | 6 | 10 |
| Other (n=4) | 3 | 1 |
| Neoadjuvant chemotherapy | | |
| Yes (n=21) | 12 | 9 |
| No (n=10) | 3 | 7 |
| Common Colorectal Cancer Mutations | | |
| KRAS (n=12) | 7 | 5 |
| MSI (n=7) | 6 | 1 |
| NRAS (n=1) | 0 | 1 |
| BRAF (n=2) | 1 | 1 |
| PIK3CA (n=2) | 2 | 0 |
| None of the above (n=11) | 2 | 9 |

**Figure 1.** Clinical characteristics of the subjects who provided the samples that created CRC patient-derived xenografts.

performing liver-specific in vivo selection with CRC PDXs, we enriched for cells optimized for growth in the liver microenvironment. Gene expression analysis revealed the gluconeogenesis enzyme PCK1 to be a robust driver of liver metastatic colonization that is over-expressed in metastatic CRC. Mechanistic studies revealed that PCK1 enhances pyrimidine nucleotide biosynthesis, which enables cancer cell growth in the context of hypoxia—a key feature of the liver microenvironment. Moreover, highly metastatic CRC PDXs contained higher levels of intermediary metabolite precursors for pyrimidine nucleotide biosynthesis. Consistent with these observations, molecular and pharmacologic inhibition of PCK1 or the pyrimidine biosynthetic gene DHODH inhibited CRC liver metastatic colonization.

## Results

### Liver growth and engraftment rates of CRC PDXs predict patient outcomes

In order to establish a PDX model of CRC liver metastatic colonization, a small sample of CRC tissue, taken either from a primary or metastatic site, was dissociated and injected subcutaneously into the flanks of NOD.Cg-*Prkdc*^scid^ *Il2rg*^tm1Wjl^/SzJ (Nod-Scid-Gamma; NSG) mice within 2 hr of surgical resection at MSKCC. Thirty-one subjects provided 40 tumor samples; 48.3% of the subjects' samples

engrafted (*Figure 1*). The majority of subjects in this study were classified as Stage IV CRC according to the American Joint Committee on Cancer (AJCC). However, AJCC stage was not associated with increased xenograft engraftment (p=0.35; $\chi^2$ test). The engraftment rates for tumor tissues that originated from the colon and the liver were similar (40% vs 37.5%; *Figure 1*). Most subjects had undergone chemotherapy prior to surgical resection of metastases (67.7%). When categorizing tumors by commonly tested clinical mutations (KRAS, high microsatellite instability (MSI-H), NRAS, BRAF, PIK3CA, and none), MSI-H tumors exhibited the highest engraftment rates (83.3%), while tumors lacking these commonly tested for clinical mutations exhibited the lowest engraftment rates (18.2%).

We found that subcutaneous tumor engraftment was associated with worse patient survival (p=0.045; *Figure 2A*). The time from subcutaneous tumor implantation to tumor harvest ranged from 35 to 88 days (*Figure 2B*). Among those CRC tumors that did grow subcutaneously, the time required to reach the pre-determined tumor size (1000 mm$^3$) was not significantly associated with patient survival (p=0.27; *Figure 2C*). When the estimated subcutaneous tumor volumes reached 1000 mm$^3$, the mice were euthanized, and the tumors were removed. For each sample, a portion of the xenografted tumor was set aside for cryopreservation, and the rest of the tumor was dissociated into a single cell suspension for portal circulation injection via the spleen. Portal circulation injection has been demonstrated to be a reliable means of establishing liver growth via hematogenous spread of CRC cells, simulating the entry of cells into the portal circulation which is typical of clinical CRC progression. After injection of cells, we observed the mice until they were deemed ill by increased abdominal girth, slow movement, and pale footpads, at which point we proceeded to euthanization and tumor extractions.

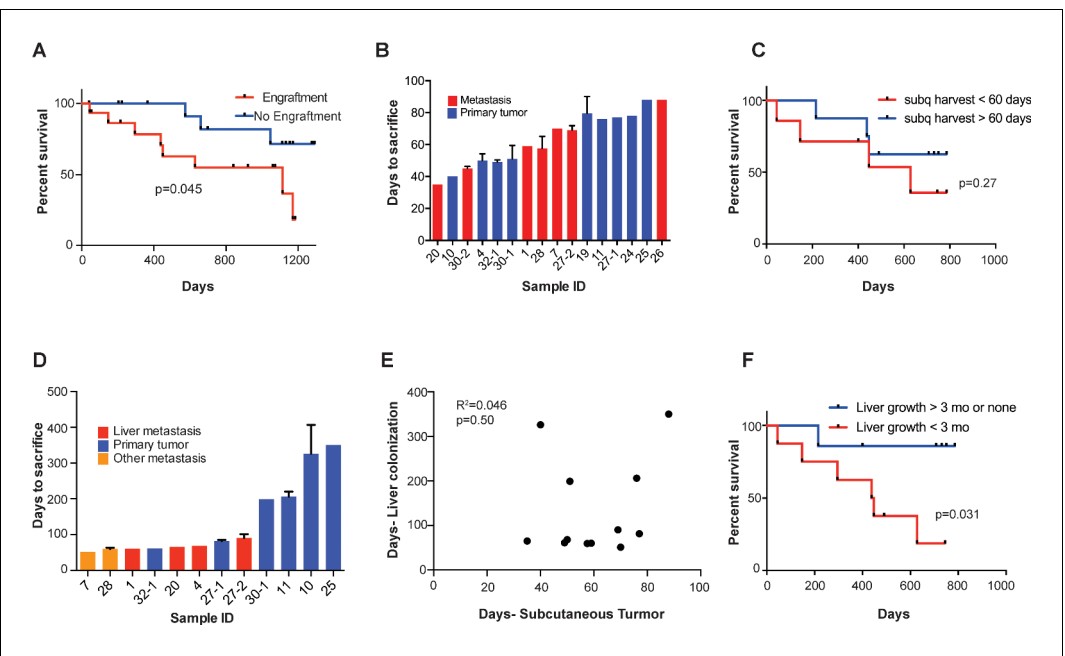

Figure 2. Subcutaneous engraftment and liver colonization growth rate, but not subcutaneous growth rate, correlate with patient outcomes in CRC PDXs. (A) Approximately 50 mm$^3$ of surgically resected colorectal cancer tissue was dissociated and injected subcutaneously into NSG mice. The mice were monitored for the presence of a subcutaneous tumor (engraftment) for 1 year. The Kaplan-Meier curve demonstrates patient survival based on subcutaneous engraftment of their corresponding CRC patient-derived xenograft (p=0.045, log-rank test). (B–C) Subcutaneous growth of CRC PDXs to the point requiring euthanasia of the host mice varied from 35 to 88 days and did not correlate with patient survival (p=0.27, log-rank test). (D–E) Subcutaneous tumor growth and liver colonization growth were not correlated (R$^2$ = 0.046, p=0.50, Pearson correlation). (F) Liver colonization growth of CRC PDXs varied from 51 to 407 days and correlated with patient survival (p=0.031, log-rank test).

The online version of this article includes the following figure supplement(s) for figure 2:

Figure supplement 1. Metastatic CRC liver PDXs fell into two biologically distinct groups.
Figure supplement 2. Histology of colorectal cancer patient-derived xenografts resembled the original tumor in both subcutaneous and liver graft sites.

Successful liver metastatic colonization was achieved upon injection of 15/17 patient samples. The time to mouse sacrifice for the CRC patient-derived liver xenografts ranged from 51 to 407 days (*Figure 2D*) and did not correlate with subcutaneous tumor growth rates ($R^2$ = 0.046, p=0.50; *Figure 2E*). The mCRC liver PDXs fell into two biologically distinct groups based on their growth rates: one set grew quickly, requiring mouse euthanasia within three months of implantation; the other set grew more slowly, requiring animal sacrifice after 6 months, or even 1-year, post-engraftment (*Figure 2—figure supplement 1A*). Importantly, these two groups of PDXs exhibited similar growth rates when implanted subcutaneously (p=0.09; *Figure 2—figure supplement 1B*). This suggests distinct selective pressures existing in the liver relative to the subcutaneous microenvironment. We found that the liver colonization model mimicked clinical outcomes, as patients whose xenografts rapidly colonized mouse livers fared poorly relative to patients whose xenografts colonized the liver slowly or not at all (p=0.031; *Figure 2F*). Taken together, these results establish that CRC liver metastasis PDX modeling described above is prognostic of clinical survival outcome.

A key objection to cell line xenografts is that the histology of animal tumors is often not representative of clinical sample histology. Contrary to this, we observed that both subcutaneous and liver engrafted tumors re-capitulated the architecture of the primary tumor from which they were derived (*Figure 2—figure supplement 2*). CLR4 was established from a poorly differentiated liver metastatic colon adenocarcinoma; it remained poorly differentiated in both the subcutaneous and liver xenografts (*Figure 2—figure supplement 2*). Similarly, CLR32 and CLR28 were derived from moderate-to-well differentiated primary colon and peritoneal metastatic adenocarcinomas and retained moderate-to-well differentiated histology when passaged subcutaneously and hepatically (*Figure 2—figure supplement 2*).

## Generation of in vivo selected highly liver metastatic PDXs

We next performed liver-directed in vivo selection through iterative splenic injections of four distinct CRC PDXs with varying mutational and metastatic backgrounds to obtain derivatives with increased capacity for liver colonization and growth (*Figure 3*). Tumors were only passaged in vivo without the use of in vitro culture. When a mouse bearing a liver colonization graft had met its pre-determined endpoint, it was euthanized, and the liver tumor was removed and dissociated into a single cell suspension in a similar manner to that of the subcutaneous tumors described above. Dissociated cells were subsequently injected into the spleen of another mouse to generate a second-generation liver metastatic derivative. This process was repeated multiple times to create a highly metastatic derivative for each of the four distinct CRC PDXs. The number of rounds of in vivo selection varied between tumor samples (range: 5–13) and in general tended to represent the number of rounds required to plateau enhanced metastatic colonization capacity. In the last round of in vivo selection, a cohort of mice was subjected to portal circulation injection with either the parental CRC PDX cells or the liver-metastatic derivative CRC PDX cells in order to assess the relative liver colonization capacities among the liver-metastatic derivatives. In each of the four CRC PDX comparisons, the in vivo-selected CRC PDX liver metastatic derivatives colonized the mouse liver more efficiently than their parental counterparts (*Figure 3*). The two extreme isogenic populations of each patient, the parental CRC PDX and its liver-metastatic derivative, were then subjected to transcriptomic and metabolite profiling as described below to identify candidate regulators of metastatic colonization.

## Candidate metastasis promoting genes identified through transcriptomic profiling of metastatic CRC PDXs

We first sought to identify candidate mCRC liver colonization promoters through mRNA sequencing and differential gene expression analyses from parental CRC PDXs (anatomical locations included subcutaneous graft, cecal graft, or first-generation liver graft) and last-generation liver-metastatic CRC PDXs. Comparisons between liver metastatic derivatives and their parental counterparts allowed for isogenic comparisons. A phylogenetic tree using complete clustering and Euclidian distance function based upon the gene expression profiles demonstrated that isogenic pairs mostly clustered together with one exception (*Figure 3—figure supplement 1A*). Using gene set enrichment analysis (GSEA) (*Subramanian et al., 2005*), we interrogated each of the four pairs of tumors individually and as a composite to identify cancer-related pathways and signatures that were significantly altered in liver metastatic derivatives compared to their isogenic parental xenografts. The hypoxia signature was found to

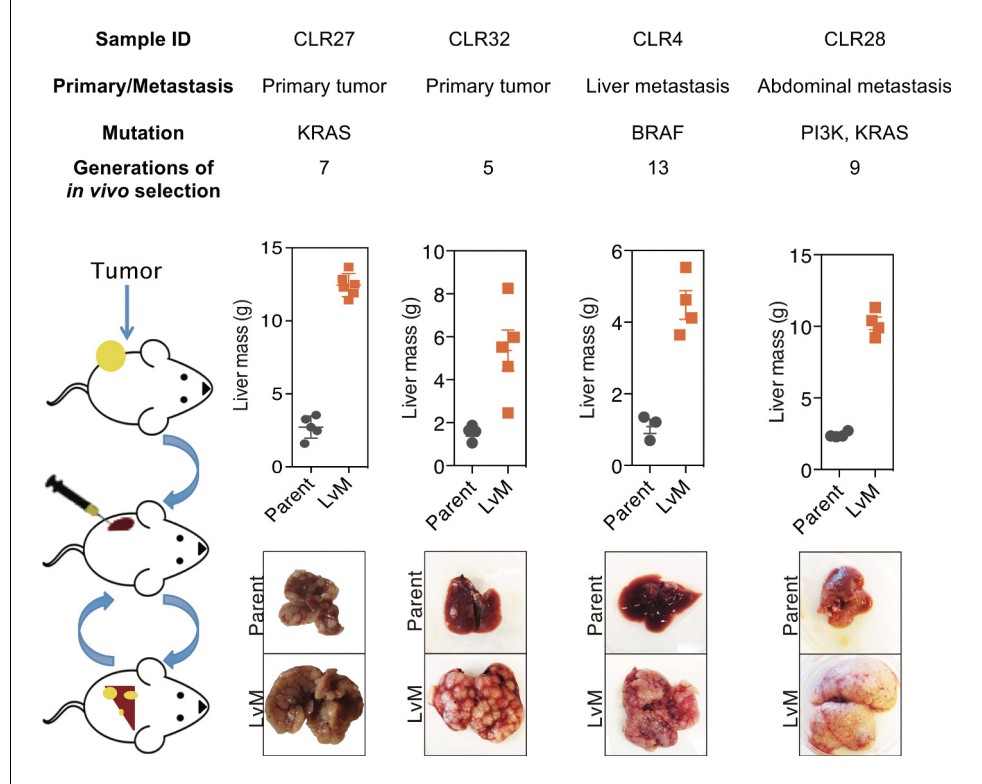

**Figure 3.** In vivo selection generates derivatives with enhanced ability to colonize mouse livers. In vivo selection was performed on four different CRC PDX samples derived from various anatomical locations and mutational backgrounds. The illustration on the left depicts the process used to generate the liver metastatic derivatives. Tumor samples from surgical specimens were inoculated subcutaneously into NSG mice. When the tumor reached the threshold size, it was removed from the mouse, dissociated into a single-cell suspension, and injected into the spleens of another set of mice as a means of introducing the colorectal cancer cells into the portal circulation. When the mice were deemed ill, the liver tumors were removed, dissociated, and re-injected to establish a next-generation liver derivative. This was repeated numerous times (Range: 5–13) with each PDX sample to obtain a distant liver metastatic CRC PDX derivative requiring euthanasia of mice in ~3 weeks after injection of cancer cells. Each of the CRC PDX liver derivatives grew significantly faster in the livers compared to their parent populations. The online version of this article includes the following figure supplement(s) for figure 3:

**Figure supplement 1.** In vivo-selected xenograft tumors more closely resembled their parental counterparts transcriptomically and led to identification of candidate metastatic liver colonization genes.

be upregulated in all the liver metastatic derivatives individually and in the composite, where it was the most significantly enriched gene signature (normalized enrichment score (NES) = 2.12, q-value <0.001; *Figure 3—figure supplement 1B*). Upregulation of hypoxia genes in the liver meta-static derivatives is consistent with previous reports demonstrating that hypoxia is an important feature of the liver metastatic microenvironment (*Loo et al., 2015*; *Nguyen et al., 2016*).

With each CRC PDX pair, we identified upregulated genes in each liver-metastatic derivative compared to its parental counterpart through a generalized linear model. The number of upregulated genes (p<0.05) in the liver-metastatic derivatives ranged from 200 (CLR28) to 345 (CLR27) out of a possible list of more than 12,000 genes. Fisher's combined probability test was used to construct a list of candidate liver colonization promoting genes that were statistically significantly upregulated across the four pairs of CRC PDXs with an effect size of greater than 1.5 log$_2$ fold change (logFC). Using this approach, we identified 24 highly upregulated genes in the liver metastatic derivatives (*Figure 3—figure supplement 1C*), with the 10 most highly upregulated genes annotated on the volcano plot (*Figure 4A*). Interestingly, two of the top ten upregulated genes (*IFITM1*, and *CKB*) have been previously implicated as promoters of CRC metastasis (*Loo et al., 2015*; *Yu et al., 2015*). The most common 'druggable' targets for cancer therapeutics are enzymes and cell-surface

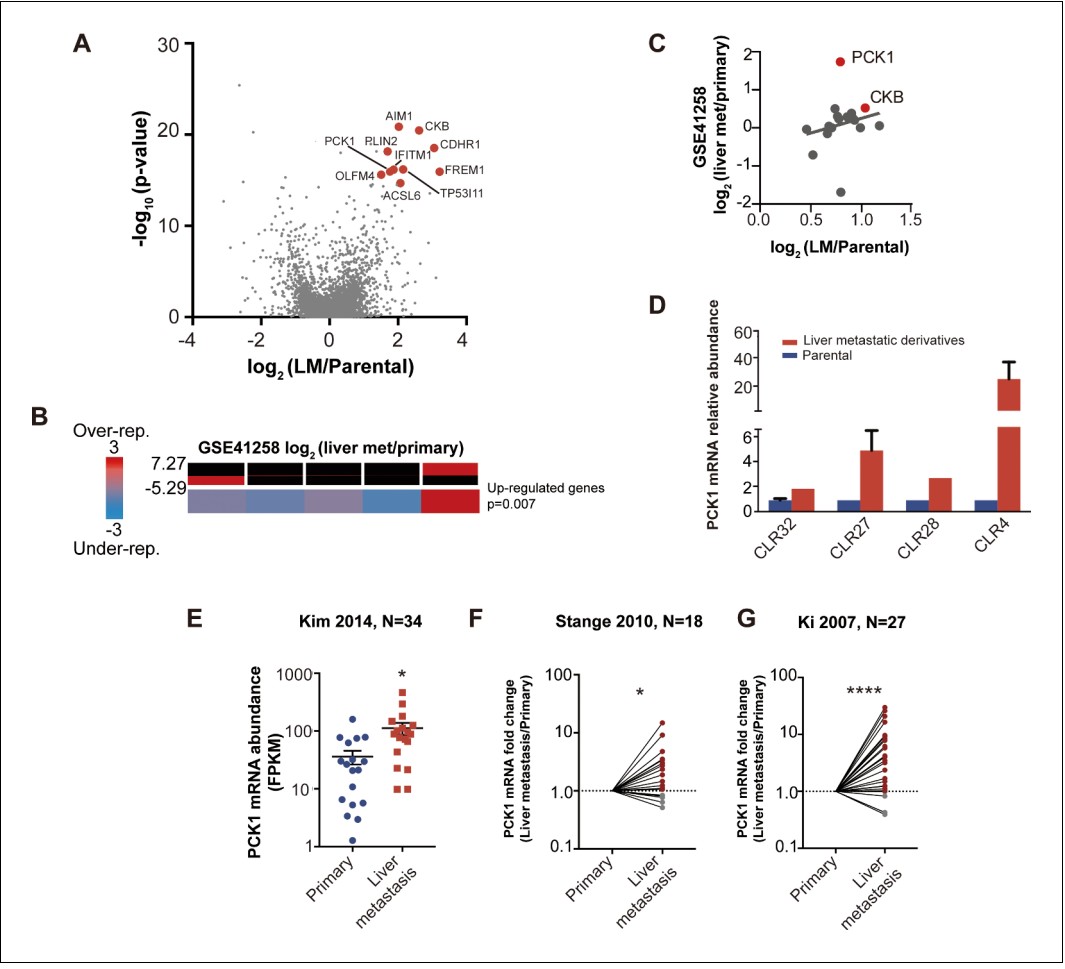

**Figure 4.** Candidate metastatic liver colonization genes identified by comparing in vivo selected liver metastatic PDXs to their parental counterparts. (**A**) Among 24 significantly upregulated genes, the 10 most highly upregulated genes (>1.5 logFC) are annotated. (**B**) The expression levels of candidate liver metastatic promoting genes were upregulated in patient liver metastases compared to primary tumors (GSE41258) (hypergeometric p=0.007). (**C**) Up-regulation of candidate liver metastasis promoting genes was significantly correlated with upregulation in patient liver metastases. Notably, *PCK1* was more upregulated in liver metastases of patients than in the mouse model (rho = 0.37, p=0.047, Pearson correlation tested with Student's t-test). (**D**) *PCK1* expression in CRC PDXs as measured by qRT-PCR. CLR32-parental (n = 3), CLR32-liver metastatic derivative, CLR27-parental, CLR27-liver metastatic derivative (n = 2), CLR28-parental, CLR28-liver metastatic derivative, CLR4-parental, and CLR4-liver metastatic derivative (n = 4). (**E**) *PCK1* is upregulated in CRC liver metastases compared to CRC primary tumors of another large publicly available dataset (GSE 50760) (p=0.01, Student's t-test). (**F–G**) *PCK1* was significantly upregulated in paired liver metastases compared to primary tumors within the same patient; this was observed in two independent datasets (GSE14297 and GSE6988) (p=0.01 in GSE14297; p<0.0001 in GSE6988, Wilcoxon matched paired signed rank test for the comparison).

receptors. In the list of candidate genes, three were enzymes (*ACSL6*, *CKB* and *PCK1*) and one was a cell-surface receptor (*CDHR1*).

One of the genes on this list, creatine kinase-brain (*CKB*), was identified by us in a prior study using established CRC cell lines and shown to act extracellularly to generate phosphocreatine, which is subsequently transported into the cell as a metabolic energetic source for ATP generation in the hypoxic microenvironment of the liver (*Loo et al., 2015*). Of the remaining three enzymes on our list, we focused on *PCK1* (phosphoenolpyruvate carboxykinase 1) given the availability of a pharmacological inhibitor and its heightened expression in normal liver (*Uhlén et al., 2015*), suggesting potential mimicry of hepatocytes by CRC cells during adaptation to the liver microenvironment.

We next investigated whether our 24-gene CRC liver colonization signature was enriched in liver metastases from patients with CRC by querying a publicly available dataset in which transcriptomes of primary CRC tumors and liver metastases were profiled. Of the 24 genes, 22 were represented in this previously published dataset (*Sheffer et al., 2009*). We binned the patient data based on differential gene expression in primary CRC tumors versus the CRC liver metastatic tumors. The upregulated genes were significantly enriched (p=0.007) in the bin with the most upregulated genes in CRC liver metastases (*Figure 4B*) (*Goodarzi et al., 2009*), supporting the clinical relevance of our in vivo-selected CRC PDX liver colonization mouse model. In further support of the clinical relevance of our findings, we found that the gene expression upregulation in our metastatic CRC system significantly correlated (rho = 0.39, p=0.047) with the gene expression upregulation in human liver CRC metastases relative to CRC primary tumors (*Figure 4C*). Interestingly, *PCK1* was highly upregulated in human CRC liver metastases relative to primary tumors. QPCR quantification confirmed *PCK1* up-regulation in liver metastatic derivatives relative to isogenic parental counterparts (*Figure 4D*). We analyzed other publicly available CRC gene expression datasets and consistently observed *PCK1* to be significantly upregulated (p=0.01, Student's t-test; *Figure 4E*) in CRC liver metastases relative to primary tumors (*Figure 4E–G*) (*Kim et al., 2014*; *Stange et al., 2010*; *Ki et al., 2007*). Additionally, *PCK1* was upregulated (p=0.01; *Figure 3F*, p<0.0001; *Figure 4G*) in CRC liver metastases in datasets containing only paired CRC primary tumors and CRC liver metastases obtained from the same patients (*Figure 4F–G*).

## *PCK1* promotes CRC liver metastatic colonization

We next performed functional in vivo studies using human CRC cell lines in which *PCK1* expression was modulated through stable gene knockdown or overexpression. Depletion of *PCK1* in SW480 cells by two independent shRNAs significantly impaired (p<0.0001 in both comparison) CRC liver metastatic colonization of cells introduced into the portal circulation of NSG mice (*Figure 5A*). *PCK1* depletion in another colorectal cell line (LS174T) also significantly decreased (p<0.0001) liver metastatic colonization (*Figure 5B*). Conversely, *PCK1* over-expression in SW480 cells significantly increased (p=0.003) liver metastatic colonization (*Figure 5C*). In contrast, *PCK1* depletion did not impact subcutaneous tumor growth in the SW480 or LS174T cell lines (*Figure 5D*). To assess whether *Pck1* modulation regulated cancer progression in a fully immunocompetent model as well, we depleted *Pck1* in the murine CRC cell line CT26. Consistent with our observations in human cancer lines, *Pck1* depletion decreased murine CRC cell liver colonization in an immune competent model (p=0.039, p=0.005 for shCTRL vs shPck1-64, shCTRL vs shPck1-66, respectively) and did not impair in vitro proliferation under basal cell culture conditions (*Figure 5E*).

We next sought to determine the cellular mechanism by which *PCK1* impacts metastatic colonization; that is, whether *PCK1* influences initial CRC cell liver colonization, apoptosis, or population growth. To identify whether initial liver colonization was the sole step in the metastatic cascade influenced by *PCK1* or whether it could provide continued impact on CRC liver growth, we generated SW480 cells expressing an inducible *PCK1* shRNA (*Figure 5—figure supplement 1A*). Four days after portal systemic injection of cancer cells, at which time CRC cells have extravasated into the liver and begun initial outgrowth, we began administering a doxycycline or a control diet. We found that even after the initial liver colonization phase (days 0–4), *PCK1* depletion continued to impair (p=0.004) CRC metastatic liver growth (*Figure 5—figure supplement 1B–C*). We did not observe increased apoptosis using the caspase 3/7 reporter in both *PCK1* depleted cell populations in vivo (*Figure 5—figure supplement 1D*). Evaluation of the in vivo growth rate through natural log slope calculations demonstrated that in each *PCK1* modulation experiment, either knockdown or overexpression, in which a luciferase reporter was used, the rate of growth after the first measured time point (day 4–7) did not equal the rate of growth of the controls (*Figure 5—figure supplement 1E*). These results reveal that *PCK1* promotes the rate of metastatic growth in vivo.

## Metabolic profiling reveals *PCK1*-dependent pyrimidine nucleotide biosynthesis in CRC under hypoxia

Given our identification of *PCK1* as a metabolic regulator of CRC liver metastatic colonization as well as the enrichment of a hypoxic signature by GSEA in highly metastatic PDXs, we speculated that perhaps *PCK1* promotes metabolic adaptation that enables growth under hypoxia—a key feature of the

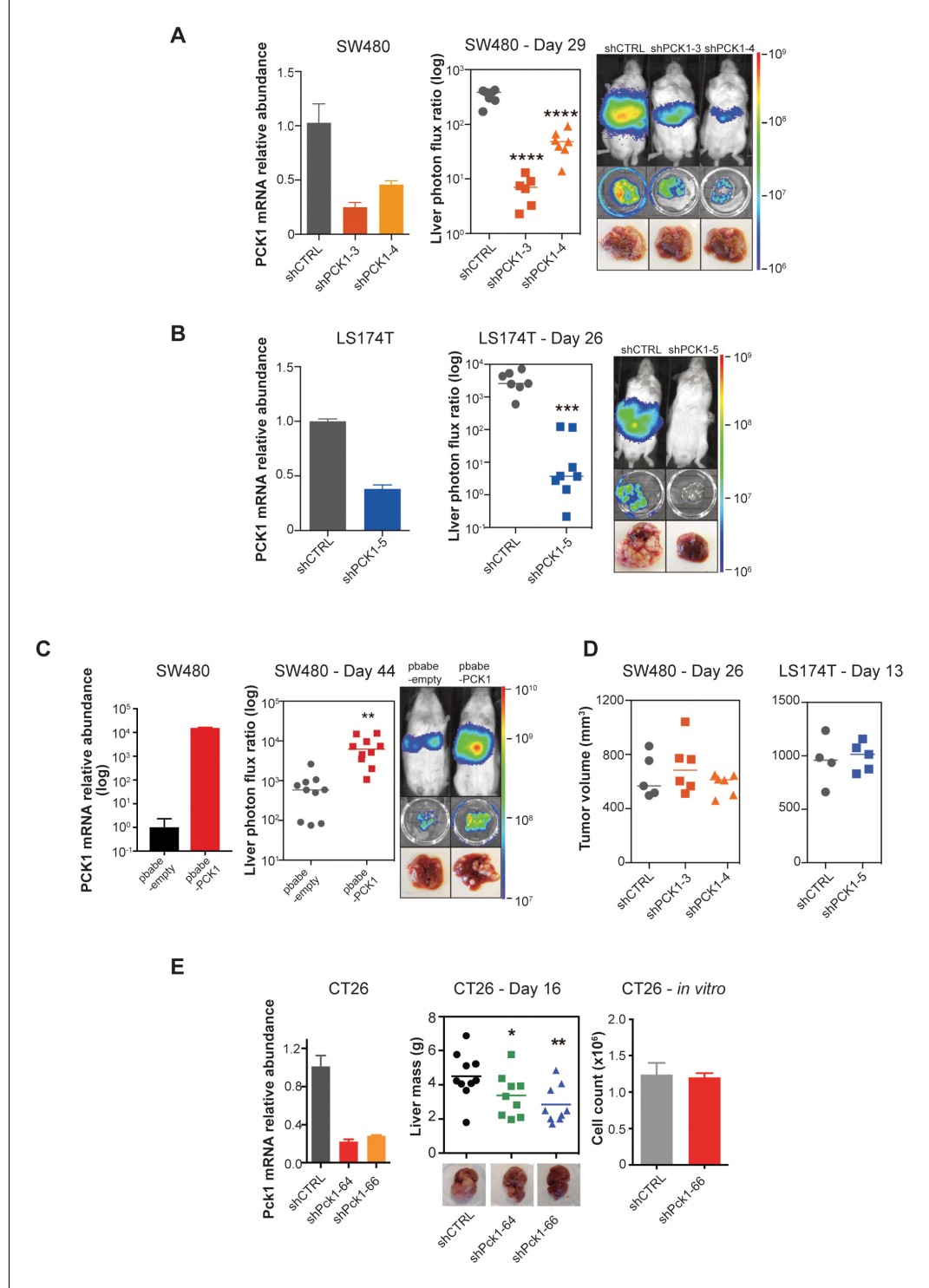

**Figure 5.** *PCK1* modulates colorectal cancer liver colonization. *PCK1* expression was measured by qRT-PCR in (**A**) *PCK1* knockdown SW480 cells (n = 3). (**B**) *PCK1* knockdown LS174T cells (n = 3) (**C**) *PCK1* overexpressing SW480 cells (n = 3), and (**E**) *PCK1* knockdown CT26 cells (n = 3). (**A**) Liver metastases in mice injected intrasplenically with 3.5 × 10⁵ SW480 cells expressing a control hairpin (n = 7) or 3.5 × 10⁵ SW480 cells expressing two independent PCK1 shRNAs (n = 6 for each shRNA hairpin) (p<0.0001 for shCTRL vs shPCK1-3 and shCTRL vs shPCK1-4, Student's t-test, Bonferroni adjusted). (**B**) Liver metastases in mice injected intrasplenically with 5 × 10⁵ LS174T cells expressing a control hairpin (n = 7) or 5 × 10⁵ LS174T cells expressing an shPCK1 (n = 8) (p<0.0001, Student's t-test). (**C**) Liver metastases in mice injected intrasplenically with 5 × 10⁴ SW480 cells expressing pbabe-empty

*Figure 5 continued on next page*

*Figure 5 continued*

control (n = 10) or $5 \times 10^4$ SW480 cells overexpressing *PCK1* (n = 10) (p=0.003, Student's t-test). (**D**) Subcutaneous tumors injected with $1 \times 10^6$ SW480 cells expressing a control hairpin (n = 5) or two independent shPCK1 hairpins (n = 6 for each shRNA hairpin) (p=0.53 for shCTRL vs shPCK1-3, p=0.45 for shCTRL vs shPCK1-4, Student's t-test); subcutaneous tumors injected bilaterally with $1 \times 10^6$ LS174T cells expressing a control hairpin (n = 4) or an shPCK1 (n = 5) (p=ns for comparison). (**E**) Liver mass of Balb-c mice injected intraspenically with $5 \times 10^5$ CT26 cells expressing either a control hairpin (n = 10) or two independent shPCK1 hairpins (n = 9 for each hairpin) (p=0.039 for shCTRL vs shPCK1-64; p=0.005 for shCTRL vs shPCK1-66, Student's t-test). In vitro growth is not affected by *PCK1* knockdown in CT26 cells (n = 4) (p=0.589, Student's t-test). $5 \times 10^4$ cells were seeded in triplicate on day 0 and counted on day 3.

The online version of this article includes the following figure supplement(s) for figure 5:

**Figure supplement 1.** *PCK1* modulation of colorectal cancer cells continuously alters population growth in the liver.

hepatic microenvironment (*Jungermann and Kietzmann, 2000*; *Dupuy et al., 2015*) (*Figure 6A*). Consistent with this, *PCK1* depletion significantly reduced CRC cell growth under hypoxia, an effect not observed under normoxia (*Figure 6B*, *Figure 6—figure supplement 1A*). Hypoxia poses a metabolic challenge for cancer cell growth as metabolites needed for biosynthesis of macromolecules required for cell proliferation can become limiting (*Birsoy et al., 2015*; *Garcia-Bermudez et al., 2018*). In vivo selected cancer cells can alter cellular metabolism in order to better respond to the metastatic microenvironment (*Loo et al., 2015*; *Nguyen et al., 2016*). To search for such adaptive metastatic metabolic alterations that associate with enhanced *PCK1* expression, we performed metabolite profiling of the four highly/poorly metastatic CRC PDX pairs. Unsupervised hierarchical clustering analysis was then performed on the differentially expressed metabolite profiles for each pair. Interestingly, the most salient observation was increased abundance in three out of four PDX pairs of multiple nucleoside base precursors and specifically metabolites in the pyrimidine biosynthetic pathway (*Figure 6C*). These metabolites comprised orotate, dihydroorotate, and ureidopropionate. These findings reveal that metastatic colonization by human CRC cells selects for induction of multiple metabolites in the pyrimidine biosynthetic pathway.

We hypothesized that perhaps enhanced levels of pyrimidine precursors were selected for in metastatic CRC cells to enable adaptation to hypoxia where precursors for pyrimidine biosynthesis such as aspartate are known to become depleted (*Birsoy et al., 2015*; *Garcia-Bermudez et al., 2018*). Without such an adaptation, cells would experience deficits in pyrimidine bases and consequently nucleotide pools, which would curb growth. How might *PCK1* upregulation contribute to maintenance of nucleotide pools? Nucleotides contain nitrogenous bases covalently coupled to ribose and phosphate. *PCK1* was previously shown to promote ribose generation by CRC cells under pathophysiological levels of glucose via the pentose phosphate pathway (*Montal et al., 2015*). We thus hypothesized that *PCK1* depletion, by depleting ribose pools, may reduce pyrimidine and purine nucleotide pools in CRC cells. To test this, we performed metabolite profiling of control and *PCK1* depleted CRC cells under hypoxia. While metabolites related to glycolysis and the citric acid (TCA) cycle were significantly increased, corroborating previous studies (*Montal et al., 2015*), the most salient finding was a significant depletion of nucleosides and nucleotides including uridine, guanine, UMP, CMP, CDP, IMP, GMP, and AMP in *PCK1* depleted cells (*Figure 6D–F*). Consistent with our observations of selective impairment of *PCK1*-dependent growth under hypoxia, the observed decreases in nucleoside and nucleotide levels were abrogated under normoxic conditions (*Figure 6—figure supplement 1B–C*). These findings reveal that increased *PCK1* expression is required for nucleotide pool maintenance in CRC cells in the context of hypoxia.

The above findings reveal that liver metastatic CRC cells enhance pyrimidine biosynthesis and that *PCK1* drives pyrimidine nucleotide synthesis under hypoxia. To better understand how *PCK1* maintains nucleotide pools under hypoxia, we performed a $^{13}$C glutamine metabolic flux analysis in control and *PCK1*-depleted cells. Glutamine labeling revealed that the generation of pyrimidine precursors, such as aspartate and orotate, by reductive carboxylation was significantly decreased upon *PCK1* depletion (*Figure 6G–H*). In contrast, the generation of pyrimidine precursors derived from oxidative reactions in the TCA cycle were either significantly increased or unchanged (*Figure 6—figure supplement 1D–E*). These findings reveal that *PCK1* promotes reductive carboxylation under

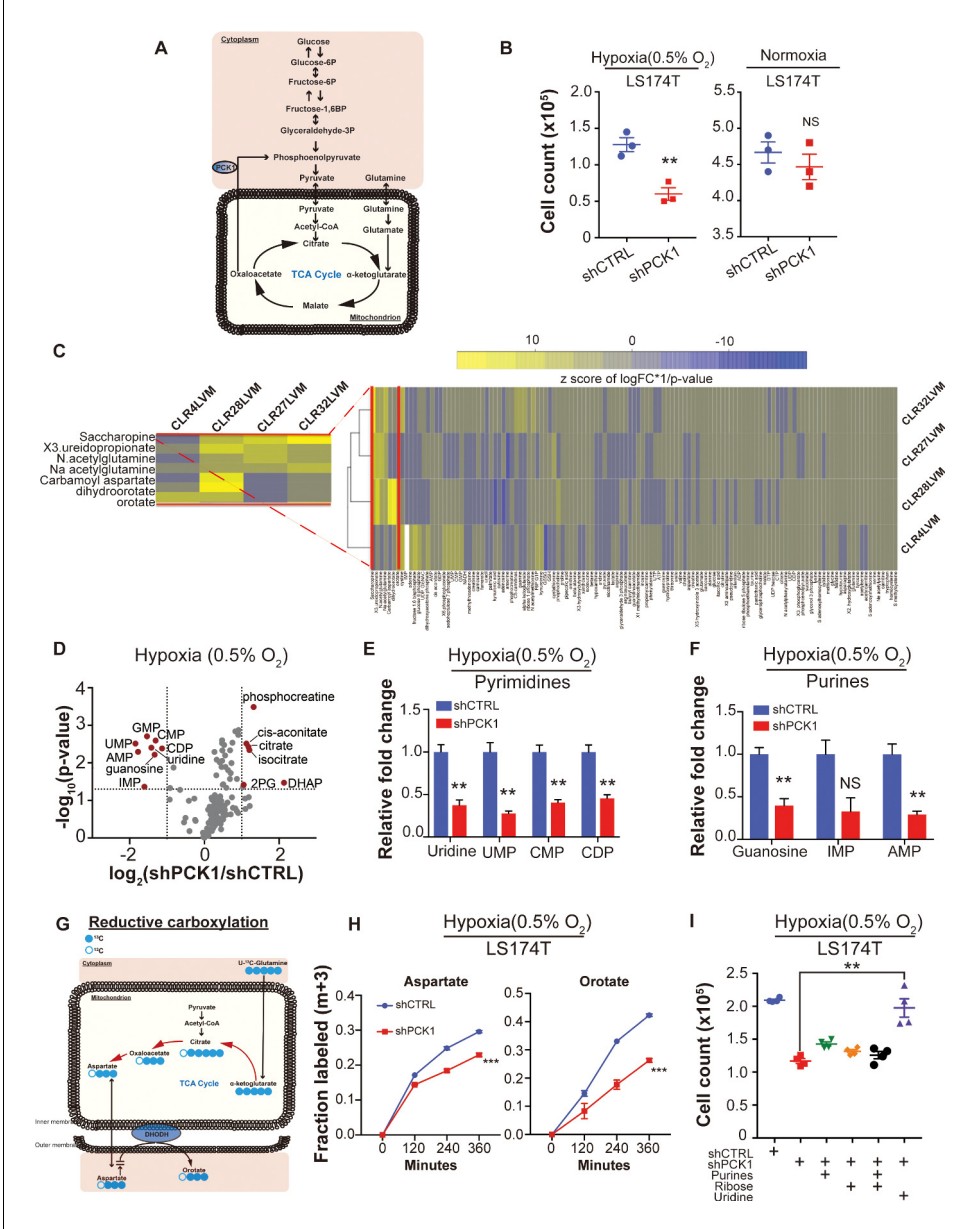

**Figure 6.** Metabolomics analysis reveals *PCK1*-dependent pyrimidine nucleotide synthesis under hypoxia in CRC. (**A**) Schematic of PCK1 related metabolic pathway. (**B**) Hypoxia and normoxia cell viability assay of LS174T cells. 2 × 10⁵ LS174T cells expressing either control hairpin or shPCK1 hairpin were cultured under normoxia for 24 hr and then were moved to hypoxic chamber (0.5% O₂) for 5 days or remained under normoxia for 5 days. *PCK1* depleted cells had a significantly lower cell count under hypoxia (p=0.006, Student's t-test). No difference under normoxia was observed (p=0.43, Student's t-test). (**C**) Unsupervised hierarchical clustering of 170 polar metabolites' profiling data. Parental PDXs were used as references to the corresponding highly metastatic PDXs. (**D**) Volcano plot showing the metabolite profile of LS174T cells expressing shCTRL or shPCK1 under hypoxia. Log₂ fold change versus -log₁₀ (p-value) was plotted. Dotted lines along x-axis represent ±log₂ (***National Cancer Institute, 2019***) fold change and dotted lines along y-axis represent -log₁₀(0.05). All metabolites either significantly enriched or depleted in shPCK1 cells compared to shCTRL are denoted as red points. All other metabolites detected are represented as gray points. (**E**) Pyrimidine metabolite levels in LS174T shCTRL versus shPCK1 cells under hypoxia. (q = 0.004, 0.003, 0.003, and 0.004 in Uridine, UMP, CMP, and CDP respectively, Student's t-test, FDR adjusted at Q value of 0.01). (**F**) Purine metabolite levels in LS174T shCTRL versus shPCK1 cells under hypoxia (q = 0.003, 0.014, and 0.003 in Guanosine, IMP, and AMP respectively, Student's t-test, FDR adjusted at Q value of 0.01). (**G**) Schematic of U-¹³C-glutamine stable isotope labeling of metabolites undergoing reductive carboxylation. (**H**) m+three fraction labeled orotate and aspartate in shCTRL or shPCK1 LS174T cells from 0 to 6 hr of U-¹³C-

*Figure 6 continued on next page*

*Figure 6 continued*

glutamine labeling under 0.5% $O_2$, m+three fraction labeled aspartate and orotate were significantly reduced in shPCK1 cells (p=0.0004 and p<0.0001 respectively, Student's t-test). (I) Hypoxia cell viability assay in 6 mM Glucose + purines, ribose, or uridine rescue. $2 \times 10^5$ LS174T cells expressing either control hairpin or shPCK1 hairpin were cultured under normoxia for 24 hr and then were moved to hypoxic chamber (0.5% $O_2$) for 5 days. Upon the exposure to hypoxia, purines (inosine and adenosine, both at 100 uM), ribose (20 mM) and uridine (100 uM) were added to the corresponding cells. (p=0.002, 0.06, 0.47, and 0.001 in shPCK1 vs shPCK1 plus purines, ribose, ribose and purines, and uridine respectively, Student's t-test, Bonferroni adjusted). All data are represented as mean ± SEM from n = 3 biological replicates.

The online version of this article includes the following source data and figure supplement(s) for figure 6:

**Source data 1.** Metabolite profiling data of shCTRL and shPCK1 expressing LS174T cells under hypoxia.
**Source data 2.** 13C glutamine flux analysis of shCTRL and shPCK1 expressing LS174T cells under hypoxia.
**Figure supplement 1.** Metabolomic analysis reveals *PCK1*-dependent pyrimidine synthesis under hypoxia in CRC.
**Figure supplement 1—source data 1.** 13C glutamine flux analysis of shCTRL and shPCK1 expressing LS174T cells under nomoxia.

hypoxia to produce pyrimidine precursor aspartate. Aspartate has been shown to be sufficient to enable growth under hypoxia or electron transport chain dysfunction (*Birsoy et al., 2015*; *Sullivan et al., 2015*). We speculated that *PCK1* might enable hypoxic growth either by promoting aspartate synthesis. To test this, we supplemented *PCK1*-depleted cells with aspartate or with pyruvate, which can i) replete NAD+ stores upon its reduction to lactate and ii) can be converted to the aspartate precursor oxaloacetate by pyruvate carboxylase in mitochondria. Supplementation of *PCK1*-depleted cells with aspartate partially rescued the observed growth defect, while pyruvate supplementation provided a modest rescue (*Figure 6—figure supplement 1F*). Interestingly, combined aspartate and pyruvate supplementation nearly fully rescued the hypoxic growth defect of *PCK1*-depleted cells (*Figure 6—figure supplement 1F*). These findings reveal that *PCK1* promotes growth by maintaining the levels of aspartate, a nucleotide precursor. The increased hypoxic growth observed upon combined aspartate and pyruvate supplementation suggest the possibility that *PCK1* may additionally promote hypoxic growth by increasing NAD+ pools.

These above findings overall suggest that hypoxia acts as a barrier to growth for metastatic CRC by limiting pyrimidine nucleoside abundance. To directly test this, we determined if the growth defect of *PCK1* depletion upon hypoxia could be rescued by the pyrimidine nucleoside uridine. Indeed, supplementation of CRC cells with uridine rescued the hypoxic growth defect caused by *PCK1* depletion. In contrast, supplementation with purines or ribose provided a modest effect (*Figure 6I*). These findings reveal *PCK1* induction to be a mechanism employed by CRC cells to generate pyrimidine nucleotide pools under hypoxia to fuel growth.

## Inhibition of *PCK1* or *DHODH* suppresses CRC liver metastatic colonization

Due to the strong reduction in mCRC liver colonization observed upon *PCK1* depletion, we hypothesized that *PCK1* inhibition may represent a potential therapeutic strategy for impairing CRC metastatic progression. We performed in vivo proof-of-principle experiments in two independent CRC cell lines with a PCK1-inhibitor, 3-mercaptopicolinic acid (3 MPA) (*DiTullio et al., 1974*). We treated CRC cells in vitro for 24 hr at a dose that did not alter cell proliferation in vitro. The following day, mice were subjected to portal circulation injections with either control or 3-MPA-treated cells. Similar to *PCK1* inhibition by shRNA, pre-treatment of cells with 3 MPA significantly reduced (p=0.01) mCRC liver colonization in vivo (*Figure 7A*, *Figure 7—figure supplement 1A*), Pre-treatment of LS174T cells with 3 MPA did not, however, alter subcutaneous tumor growth (*Figure 7—figure supplement 1B*). We next sought to determine whether experimental therapeutic delivery of 3 MPA could suppress metastatic colonization. Prior to portal systemic injection of LS174T cells, we began oral gavage treatment of mice with either 200 mg/kg of 3 MPA in aqueous solution or control. On day 1, we repeated the 3 MPA or control gavage. We found that even such short-term treatment of 3 MPA decreased CRC liver colonization in this model (*Figure 7B*, *Figure 7—figure supplement 1C*) (p=0.008; p=0.005 for *Figure 7B* and *Figure 7—figure supplement 1C* respectively). Taken

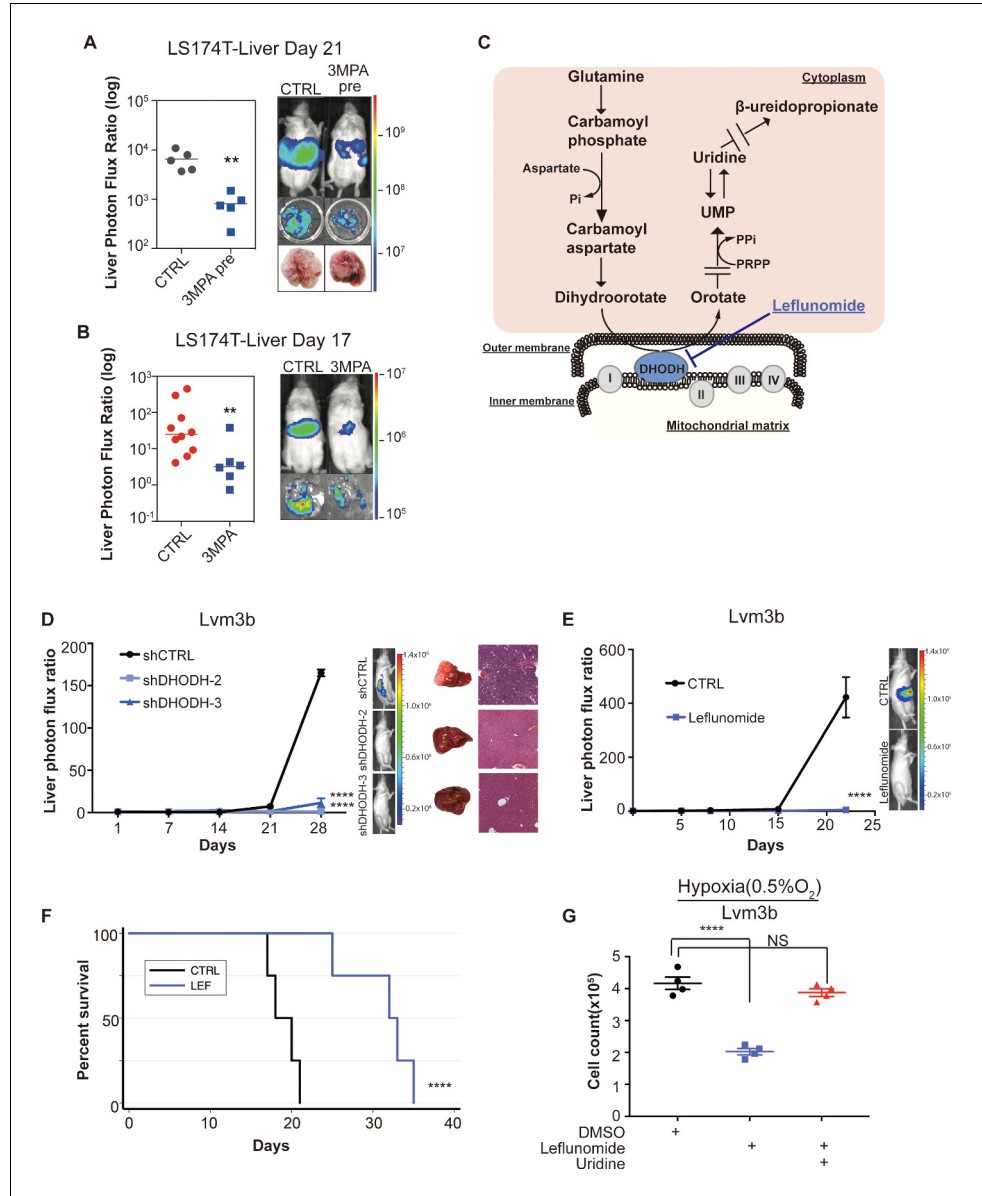

**Figure 7.** Genetic and pharmacologic inhibition of *DHODH* decreases in vivo CRC liver metastatic colonization. (**A**) 3 MPA pre-treatment decreased liver colonization of LS174T CRC cells. $1 \times 10^6$ control LS174T cells or 3 MPA pre-treated LS174T cells were injected intrasplenically (p=0.01, Student's t-test). (**B**) Oral administration of 3 MPA decreased liver colonization of LS174T CRC cells. One hour prior to the intrasplenic injection of $1 \times 10^6$ LS174T cells, mice were gavaged with either 3 MPA (200 mg/kg in aqueous solution; n = 6) or control (n = 10). Mice received a second dose on day 1. Four mice in the 3 MPA died prior to imaging on day 3 and were not included in the analysis. two independent experimental results were pooled. (p=0.008 for 3 MPA vs control, Student's t-test). (**C**) Schematic of de novo pyrimidine synthesis pathway. (**D**) Genetic silencing of *DHODH* decreased liver colonization of mCRC cells. $1 \times 10^6$ Lvm3b cells expressing a control hairpin or hairpins targeting *DHODH* were intrasplenically injected to athymic nude mice (n = 4 per each cohort) on day 1 and were imaged every week. Luciferase bioluminescent images and corresponding gross liver images and H and E stains are depicted (p<0.0001 in shDHODH-2 and p<0.0001 in shDHODH-3, Student's t-test, Bonferroni adjusted). (**E**) Leflunomide inhibits liver metastatic colonization of Lvm3b cells. $1 \times 10^6$ Lvm3b cells were intrasplenically injected into athymic nude mice (n = 4 per each cohort) on day 1 and leflunomide (7.5 mg/kg mouse body weight) or DMSO treatment was begun on day 1. The mice were imaged every week. Firefly luciferase bioluminescent images are shown (p<0.0001, Student's t-test). (**F**) Kaplan-Meier plot of the experiment shown in B (n = 4 per each cohort) (p=0.007, log-rank test). (**G**) Uridine supplementation rescued leflunomide induced cell growth reduction under hypoxia. $2 \times 10^5$ Lvm3b cells were cultured under normoxia for 24 hr and then were moved to hypoxic chamber (0.5% $O_2$) for 5

*Figure 7 continued on next page*

*Figure 7 continued*

days. Upon the exposure to hypoxia, leflunomide (100 uM) and uridine (100 uM) was added to the corresponding cells (p<0.001 for DMSO vs leflunomide; p=0.25 for DMSO vs leflunomide/uridine, Student's t-test).
The online version of this article includes the following figure supplement(s) for figure 7:

**Figure supplement 1.** *DHODH* inhibition suppressed metastatic liver colonization and hypoxic cell growth of CRC cells.

together, these results indicate that *PCK1* promotes colorectal cancer liver colonization and represents a potential therapeutic target in the prevention of CRC at risk for metastatic disease.

Dihydroorotate dehydrogenase is a key enzyme in the metabolic pathway that reduces dihydroorotate to orotate, which is ultimately converted to the pyrimidine nucleotides UTP and CTP (*Figure 7C*). To further confirm that pyrimidine biosynthesis promotes CRC hypoxic growth, we sought to assess CRC growth upon DHODH inhibition. Leflunomide is an approved, well-tolerated, and high-affinity (Kd = 12 nM) small-molecule inhibitor of DHODH used in the treatment of rheumatoid arthritis. Leflunomide treatment significantly impaired CRC growth in the context of hypoxia—an effect that was more significant under hypoxia than normoxia (*Figure 7—figure supplement 1D–E*). These findings confirm that metastatic CRC cell growth under hypoxia is sensitive to pyrimidine biosynthesis inhibition.

Our findings as a whole suggest that metastatic CRC liver metastatic colonization may be sensitive to inhibition of the pyrimidine biosynthetic pathway. To directly test this, we first depleted highly metastatic Lvm3b CRC cells of *DHODH* (*Figure 7—figure supplement 1F*). *DHODH* depletion substantially reduced CRC liver metastatic colonization (*Figure 7D*), revealing a critical role for *DHODH* activity and pyrimidine biosynthesis in CRC liver metastatic colonization. To determine if leflunomide can therapeutically inhibit CRC liver metastasis, we treated animals with a dose of this drug similar to that used for rheumatoid arthritis (7.5 mg/kg body weight). Treatment of animals injected with highly metastatic Lvm3b cells with leflunomide caused a ~ 90 fold reduction in CRC liver metastatic colonization (*Figure 7E*). The leflunomide treated mice experienced significantly longer survival (p=0.006) than the control mice (*Figure 7F*). Importantly, these cells are known to be highly resistant to 5-FU (*Bracht et al., 2010*), the backbone chemotherapeutic used in CRC, revealing that inhibition of *DHODH* can exert therapeutic benefit despite cellular resistance to an anti-metabolite that targets the pyrimidine pathway. Leflunomide treatment only modestly impacted primary tumor growth by two distinct CRC populations (*Figure 7—figure supplement 1G–H*), suggesting enhanced sensitivity of CRC cells to leflunomide-mediated DHODH inhibition during liver metastatic colonization. To determine if the metastatic colonization defect caused by leflunomide treatment is caused by pyrimidine depletion, we tested cell growth suppression in the presence or absence of uridine—the downstream metabolic product of the pyrimidine biosynthetic pathway. This revealed that the impaired growth upon hypoxia was rescued upon uridine supplementation (*Figure 7G*). Importantly, leflunomide treatment impaired proliferation significantly more in the context of hypoxia than under normoxia (*Figure 7—figure supplement 1D–E*). These observations reveal enhanced dependence of highly metastatic cells on pyrimidine biosynthesis and reveal upregulation of metabolites in this pathway as a selective adaptive trait of highly metastatic CRC cells. Overall, these findings identify *DHODH* as a therapeutic target in CRC progression and provide proof-of-concept for use of leflunomide in therapeutic inhibition of CRC metastatic progression.

## Discussion

CRC remains a challenging disease despite multiple advances over the last six decades. Some patients with metastatic CRC can experience regression responses to current therapies, although most succumb to their disease within 3 years. Given that most CRC deaths occur as a result of complications of metastatic disease, a model that can predict which patients with advanced CRC harbor more aggressive disease could aid in appropriately positioning patients for experimental clinical trials. The objectives of our study were two-fold: to develop a CRC liver metastasis patient-derived xenograft model, and to employ this model to identify candidate genes and biology underlying CRC liver metastatic colonization.

Most patient-derived xenograft models consist of subcutaneous tumor tissue implantation. Similar to others, we found that successful subcutaneous tumor engraftment associated with worse patient survival in those with CRC (*Oh et al., 2015*). However, among those tumors in our study that did engraft subcutaneously, the subcutaneous tumor growth rate did not significantly correlate with patient survival. In contrast, we found that liver metastatic growth rate was significantly correlated with patient survival. The reason for this discrepancy in the prognostic power of subcutaneous tumor growth versus liver metastatic growth is likely the greater selective pressure inherent to the liver microenvironment. This collection of clinically predictive CRC liver metastatic PDX models represents a valuable resource for the cancer community.

*PCK1* is the rate-limiting enzyme in gluconeogenesis and is often upregulated in patients with metabolic syndrome and diabetes mellitus. Epidemiologic data suggests that those patients with diabetes that are on metformin, a gluconeogenic-antagonist, exhibit improved CRC clinical outcomes relative to their metformin-free counterparts (*Zhang et al., 2011*; *Spillane et al., 2013*; *Fransgaard et al., 2016*). Our observations suggest a potential mechanistic basis for the sensitivity of CRC metastatic progression to inhibition of this metabolic pathway.

Metabolic rewiring in cancer has been well-established to provide tumor cells with the necessary nutrients and anabolic components to sustain proliferative and energetic demands (*Chandel, 2015*; *Vander Heiden and DeBerardinis, 2017*; *Boroughs and DeBerardinis, 2015*). While numerous pathways are involved in metabolic reprogramming, metabolic shunting into pathways including glucose metabolism, the citric acid (TCA) cycle, and lipogenesis largely support macromolecule synthesis for cancer cells (*Locasale and Cantley, 2011*; *Vander Heiden et al., 2009*; *DeBerardinis et al., 2008*; *Possemato et al., 2011*). In line with these notions, there have been two reports on *PCK1* and its role in cancer. In one study, *PCK1* was found to enhance melanoma tumor re-initiation (*Li et al., 2015*). In this work, the authors demonstrated that, in tissue culture, melanoma 'tumor re-initiating cells' consumed more glucose and produced more lactate and glycerate-3-phosphate; *PCK1* silencing elicited the opposite phenotype in culture (*Li et al., 2015*). Using cell culture metabolomics, *Montal et al. (2015)* recently described a mechanism by which *PCK1* promotes CRC growth through its increased ability to metabolize glutamine into lipids and ribose . *PCK1* silencing in a CRC cell line in vitro was shown to decrease glutamine utilization and TCA cycle flux (*Montal et al., 2015*). This group found that cells with increased expression of *PCK1* consumed more glucose and produced more lactate. They further performed *PCK1* staining on a primary CRC tissue microarray, finding that *PCK1* was overexpressed in many primary CRC biopsies but its expression was not associated with tumor grade. Our findings demonstrate a major role for *PCK1* in liver metastatic colonization by CRC. While we did not find evidence of *PCK2* upregulation in our mCRC model, three recent studies demonstrated that *PCK2* upregulation in lung cancer cells in vitro can improve cancer cell survival in glucose-depleted conditions (*Leithner et al., 2015*; *Vincent et al., 2015*). Vincent et al. found that in glucose-depleted conditions, lung cancer cells increased consumption of glutamine as an energy source in a PCK2-dependent manner. *Zhao et al. (2017)* observed *PCK2* upregulation in tumor initiating cells and demonstrated that *PCK2* promoted tumor initiation through reducing TCA cycle flux by lowering Acetyl-CoA. Our findings provide three novel insights underlying the role of *PCK1* in cancer progression. First, we demonstrate that increased *PCK1* strongly drives liver metastatic colonization but minimally impacted primary tumor growth rate in the cells studied. Second, we provide the first reported evidence that *PCK1* can promote hypoxic growth. Third, we uncover a key role for *PCK1* in pyrimidine biosynthesis under hypoxia via reductive carboxylation. Although we do not provide a precise biochemical mechanism by which *PCK1* promotes reductive carboxylation, *Zamboni et al. (2004)* reported that in a bacterium with pyruvate kinase impairment, *PCK1* could operate in reverse—converting phosphoenolpyruvate to oxaloacetate. We speculate that such a reaction would drive reductive carboxylation by generating oxaloacetate, a precursor for aspartate and consequently pyrimidines. In support of this, our metabolite profiling of *PCK1*-depleted cells under hypoxia revealed increased abundance of phosphoenolpyruvate. Moreover, we had previously observed that highly metastatic CRC cells upregulate *PKLR*, which was shown to inhibit pyruvate kinase M2 (*PKM2*) activity (*Nguyen et al., 2016*)—analogous to the observations in the aforementioned bacterial study. Further investigation is warranted to conclusively define the mechanism by which *PCK1* regulates pyrimidine synthesis in metastatic CRC cells under hypoxia.

Because tumor cells alter their metabolic programs within their given tumor microenvironment, metabolic liabilities that provide therapeutic opportunities emerge (*Li et al., 2015*; *Wheaton et al., 2014*; *Locasale et al., 2011*). Recent work has implicated *DHODH* as a regulator of differentiation in myeloid leukemia as well as a promoter of cell cycle progression in some solid cancer types such as melanoma and pancreatic adenocarcinoma (*White et al., 2011*). Bajzikova et al. found that de-novo pyrimidine biosynthesis is essential for mouse breast cancer tumorigenesis in a *DHODH*-dependent manner (*Bajzikova et al., 2019*). Our work reveals that beyond effects on cell growth in vitro and primary tumor growth, CRC metastatic progression selects for upregulation of pyrimidine biosynthesis. Moreover, the use of leflunomide to therapeutically target *DHODH* has been implicated under various cancer contexts as a metabolic inhibitor (*Mathur et al., 2017*; *Luengo et al., 2017*). Here, we observe that molecular or pharmacological inhibition of this pathway with leflunomide strongly impairs CRC metastatic colonization relative to primary tumor growth. Our work reveals that hypoxia enhances the sensitivity of cells to DHODH inhibition, consistent with enhanced pyrimidine biosynthesis enabling enhanced growth under hypoxia—a key feature of the hepatic tumor microenvironment.

5-Fluorouracil (5-FU) was the first chemotherapeutic to demonstrate efficacy in reducing the risk of CRC recurrence (*Moertel et al., 1990*). This agent remains the backbone of the current FOLFOX regimen, which is administered to patients after surgical resection to reduce the risk of metastatic relapse. Interestingly, 5-FU targets thymidylate synthase, an enzyme downstream of *DHODH* in the pyrimidine biosynthetic pathway—supporting our premise of dependence and susceptibility to inhibition of this pathway in CRC metastasis. Despite its activity, a large fraction of patients treated with 5-FU relapse. Multiple mechanisms of resistance to 5-FU have been described (*Holohan et al., 2013*). Our findings demonstrate that inhibition of *DHODH* can suppress metastatic progression of a CRC cell line that is resistant to 5-FU—revealing promise for clinical testing of this agent in patients at high risk for relapse and whose tumors may exhibit resistance to 5-FU. Overall, our work reveals that PDX modeling of CRC can be predictive of clinical survival outcomes; that integration of PDX modeling with in vivo selection can give rise to highly metastatic PDX derivatives which can be profiled transcriptomically and metabolically to identify key drivers of metastatic progression; and that *PCK1* and *DHODH* represent key metabolic drivers of CRC metastasis and therapeutic targets in CRC.

# Materials and methods

## Key resources table

| Reagent type (species) or resource | Designation | Source or reference | Identifiers | Additional information |
|---|---|---|---|---|
| Gene (*Homo sapiens*) | *PCK1* | | HGNC: 8724 | |
| Gene (*M. musculus*) | *Pck1* | | MGI:97501 | |
| Gene (*Homo sapiens*) | *DHODH* | | HGNC: 2867 | |
| Cell line (*Homo sapiens*) | SW480 | ATCC | CCL-228 | RRID:CVCL_0546 |
| Cell line (*Homo sapiens*) | LS174T | ATCC | CL-188 | RRID:CVCL_1384 |
| Cell line (*Homo sapiens*) | Lvm3b | Tavazoie Lab | | Derived from in vivo selection of LS174T cells |
| Cell line (*Homo sapiens*) | 293LTV | Cell Biolabs | LTV100 | RRID: CVCL_JZ09 |

*Continued on next page*

*Continued*

| Reagent type (species) or resource | Designation | Source or reference | Identifiers | Additional information |
|---|---|---|---|---|
| Cell line (*M. musculus*) | CT26 | ATCC | CRL-2638 | RRID: CVCL_7256 |
| Transfected construct (*Homo sapiens*) | shRNA to PCK1 | Sigma | PCK1 sh3 (TRCN0000196706), PCK1 sh4 (TRCN0000199286), PCK1 sh5 (TRCN0000199573), shControl (SHC002), | Lentiviral construct to transfect and express the shRNA. |
| Transfected construct (*Homo sapiens*) | shRNA to DHODH | Sigma | DHODH sh2 (TRCN0000221421), and DHODH sh3 (TRCN0000221422) | Lentiviral construct to transfect and express the shRNA. |
| Biological sample (*Homo sapiens*) | Colorectal cancer PDXs | MSKCC | CLR1, 3, 4, 7, 20,27–1, 28, 32–1, 10, 11, 19, 24, 25, 26, and 30 | MSKCC Institutional Review Board/Privacy Board (protocol 10-018A) |
| Biological sample (*Homo sapiens*) | In vivo selected colorectal cancer PDXs | Tavazoie Lab | CLR4LVM, CLR27LVM, CLR28LVM, and CLR32LVM | The Rockefeller University Institutional Review Board (protocol STA-0681) |
| Antibody | APC-anti human CD326(mouse monoclonal) | BioLegend | 324207 | FACS: 5 ul/ 1 million cells RRID:AB_756081 |
| Antibody | FITC-anti mouse H-2K$^d$(mouse monoclonal) | BioLegend | 116606 | FACS: 1 ul/ 1 million cells RRID:AB_313741 |
| Recombinant DNA reagent | pLKO.1-puro | Addgene | 8453 | RRID: Addgene_8453 |
| Recombinant DNA reagent | pBABE-puro | Addgene | 1764 | RRID: Addgene_1764 |
| Recombinant DNA reagent | plx304-blast | Addgene | 25890 | RRID: Addgene_25890 |
| Sequence-based reagent | PCK1-F (*Homo sapiens*) | This paper | qPCR primer | AAGGTGTTCCCATTGAAGG |
| Sequence-based reagent | PCK1-R (*Homo sapiens*) | This paper | qPCR primer | GAAGTTGTAGCCAAAGAAGG |
| Sequence-based reagent | PCK1-F (*M. musculus*) | This paper | qPCR primer | CTGCATAACGGTCTGGACTTC |
| Sequence-based reagent | PCK1-R (*M. musculus*) | This paper | qPCR primer | CAGCAACTGCCCGTACTCC |
| Commercial assay or kit | MACS kit | Miltenyi | 130-104-694 | |
| Chemical compound, drug | Leflunomide | Tocris | 2228 | |
| Software, algorithm | PRISM | Graphpad | Version 8 | |
| Software, algorithm | Rstudio | Rstudio, Inc. | Version 1.2.5001 | |
| Other | U-$^{13}$C-glutamine | Cambridge Isotope Laboratories | CLM-1822-H | |

## Cell culture

SW480, LS174T, and CT26 cell lines were obtained from ATCC. HEK-293LTV cells were obtained from Cell Biolabs. LS174T, HEK-293LTV, and CT26 cells were grown in Dulbecco's Modified Eagle Medium (Gibco) supplemented with 10% v/v fetal bovine serum (Corning), L-glutamine (2 mM; Gibco),

penicillin-streptomycin (100 U/ml; Gibco), Amphotericin (1 μg/ml; Lonza), and sodium pyruvate (1 mM; Gibco). SW480 cells were grown in McCoy's 5A modified media with L-glutamine (Corning) supplemented with 10% v/v fetal bovine serum, penicillin-streptomycin (100 U/ml), Amphotericin (1 μg/ml), and sodium pyruvate (1 mM). All cells were grown at 37°C under 5% $CO_2$ and passaged when the monolayer reached 80% confluency. All cell lines were authenticated by SPR profiling at MSKCC. All cells were regularly checked for mycoplasma contamination and have been negative.

## In vitro cell growth assays

CT26 cells that had been stably transduced with *PCK1*-targetting shRNA hairpins or control hairpins were grown in vitro for 3 days and counted on day three using the Sceptor 2.0 automated Cell counter (Millipore).

## In vitro hypoxia cell growth assays

Lvm3b cells or LS174T cells were grown under normoxia for 24 hr followed by incubation for 5 days under 0.5% oxygen and then counted using the Sceptor 2.0 automated Cell Counter (Millipore).

## 3-Mercaptopicolinic acid in vitro growth assay

LS174T cells were seeded in 6-well plates. On day 1, the media was replaced with either control media or media supplemented with 1 mM 3 MPA. On day 2, all the media was replaced with control media. The experiment was terminated on day 5.

Twenty-four hours exposure to 1 mM 3 MPA in media does not alter LS174T cell growth in vitro. $2 \times 10^4$ LS174T cells were seeded in triplicate. On day 1, the media was replaced with either control media or media supplemented with 1 mM 3 MPA. On day 2, all the media was replaced with control media. The experiment was terminated on day 5.

## Stable cell lines

Lentiviral particles were created using the ViraSafe lentiviral packaging system (Cell Biolabs). ShRNA oligo sequences were based upon the Sigma-Aldrich MISSION shRNA library and were obtained from Integrated DNA technologies. The following shRNAs were used in this study: PCK1 sh3 (TRCN0000196706), PCK1 sh4 (TRCN0000199286), PCK1 sh5 (TRCN0000199573), shControl (SHC002), mouse PCK1 sh64 (TRCN0000025064), mouse PCK1 sh66 (TRCN0000025066), DHODH sh2 (TRCN0000221421), and DHODH sh3 (TRCN0000221422). Forward and reverse complement oligos were annealed, cloned into pLKO, and transformed into XL10-Gold *E. coli* (200314, Agilent). For PCK1 overexpression, PCK1 cDNA (plasmid ID HsCD00045535) was obtained from the PlasmID Repository at Harvard Medical School and cloned into pBabe-puromycin or plx304-blasticidin. For tetracycline-inducible experiments, the seed sequences of shRNA control (SHC002) or PCK1 sh4 (TRCN0000199286) were cloned into pLKO-Tet-On (*Wiederschain et al., 2009*). All plasmids were isolated using the plasmid plus midi kit (Qiagen). Transduction and transfection were performed as described previously (*Yu et al., 2015*).

## Animals (studies)

All animal works were conducted in accordance with a protocol approved by the Institutional Animal Care and Use Committee (IACUC) at The Rockefeller University and Memorial Sloan Kettering Cancer Center. Either NOD.Cg-Prkdc$^{scid}$ Il2rg$^{tm1Wjl}$/SzJ (Nod-Scid-Gamma; NSG) aged 6–10 weeks or Foxn1$^{nu}$(Nu/J; athymic nude) aged 6–10 weeks were used for all mouse experiments. For functional studies of PCK1, CRC cells (SW480 or LS174T) that had been stably transduced with a luciferase reporter (*Ponomarev et al., 2004*) were subjected to portal circulation injection in NSG mice; after two minutes, a splenectomy was performed. For functional and pharmacology studies of DHODH, CRC cells (Lvm3b) that had been stably transduced with a luciferase reporter were subjected to portal circulation injection in athymic nude mice; after 2 min, a splenectomy was performed. Mice were imaged weekly; experiments were terminated when the luciferase signal had saturated or the mice were too ill, whichever occurred first.

## Administration of 3-Mercaptopicolinic acid and leflunomide in vivo

Chow was removed from cages four hours prior to injection of LS174T cells. Gavage with either 3 MPA (200 mg/kg in aqueous solution) or placebo was performed one hour prior to injection of LS174T cells. $1 \times 10^6$ LS174T cells were injected into the portal systemic circulation as described *Animals* section above. Chow was returned to cages after injection. On day 1, chow was removed from cages; 3 MPA or placebo was administered via gavage 4 hr post-chow removal. Chow was returned to cages four hours after drug administration. Mice were imaged bi-weekly. Leflunomide (Tocris Cat # 2228) 7.5 mg/kg mouse body weight was intraperitoneally injected every day. Equivalent volume of DMSO were intraperitoneally injected every day to the control cohort.

## Histology

Patient colorectal tumors were prepared and stained with hemotoxylin and eosin (H and E) per standard clinical procedures following surgical resection of the tumor specimen. Subcutaneous and liver xenograft samples were removed from the mice at time of sacrifice and fixed in 4% paraformaldehyde solution for 48 hr at 4°. The xenografts samples were subsequently rinsed in PBS twice followed by one-hour incubations in 50% ethanol, then 70% ethanol. The xenografts samples were stained with maintained in 70% ethanol at 4°. The fixed xenografts samples were embedded in paraffin, sectioned, and stained with H and E (Histoserv).

Quantitative RT-PCR qRT-PCR was performed to confirm expression of PCK1. Total RNA was extracted (37500, Norgen) from CRC PDXs, SW480, LS174T, or CT26 cells that had been stably transduced with PCK1-targetting shRNA hairpins, control hairpins, pBabe-PCK1, pBabe-control, plx-PCK1, or plx-Empty. cDNA was generated using Superscript III first strand cDNA synthesis kit (18080051, Invitrogen) per manufacturer's protocol. For quantification of cDNA, Fast SYBR Green Master Mix (4385612, Applied Biosystems) was used for sample analysis. Gene expression was normalized to HPRT expression. The following sequences were used as primers for CRC PDXs, SW480, and LS174T cells: PCK1-F, AAGGTGTTCCCATTGAAGG; PCK1-R, GAAGTTGTAGCCAAAGAAGG; HPRT-F, GACCAGTCAACAGGGGACAT; HPRT-R, CCTGACCAAGGAAAGCAAAG. The following sequences were used as primers for CT26 cells: PCK1-F, CTGCATAACGGTCTGGACTTC; PCK1-R, CAGCAACTGCCCGTACTCC; b-actin-F, GGCTGTATTCCCCTCCATCG; b-actin-R, CCAGTTGGTAA-CAATGCCATGT. The following primers were used for Lvm3b cells: DHODH-F, CCACGGGAGA TGAGCGTTTC; DHODH-R, CAGGGAGGTGAAGCGAACA.

## Clinical analysis

GEO data sets GSE41258, GSE 507060, GSE14297, and GSE6988 were used to evaluate for expression of *PCK1* as described previously (*Yu et al., 2015*; *Kim et al., 2014*).

## Patient-derived colorectal cancer xenografts

Within 2 hr of surgical resection, CRC tumor tissue that was not needed for diagnosis was implanted subcutaneously into NSG mice at the MSKCC Antitumor Assessment Core facility. When the tumor reached the pre-determined end-point of 1,000 mm$^3$, the tumor was excised and transferred to the Rockefeller University. Xenograft tumor pieces of 20–30 mm$^3$ were re-implanted. When the subcutaneous tumor reached 1,000 mm$^3$, the tumor was excised. Part of the tumor was cryogenically frozen in FBS:DMSO (90:10) for future use. The rest of the tumor was chopped finely with a scalpel and placed in a 50 ml conical tube with a solution of Dulbecco's Modified Eagle Medium (Gibco) supplemented with 10% v/v fetal bovine serum (Corning), L-glutamine (2 mM; Gibco), penicillin-streptomycin (100 U/ml; Gibco), Amphotericin (1 µg/ml; Lonza), sodium pyruvate (1 mM; Gibco) and Collagenase, Type IV (200 U/ml; Worthington) and placed in a 37°C shaker at 220 rpm for 30 min. After centrifugation and removal of supernatant, the sample was subjected to ACK lysis buffer (Lonza) for 3 min at room temperature to remove red blood cells. After centrifugation and removal of ACK lysis buffer, the sample was subjected to a density gradient with Optiprep (1114542, Axis-Shield) to remove dead cells. The sample was washed in media and subjected to a 100-µm cell strainer and followed by a 70-µm cell strainer. Mouse cells were removed from the single-cell suspension via magnetic-associated cell sorting using the Mouse Cell Depletion Kit ((130-104-694, Miltenyi), resulting in a single-cell suspension of predominantly CRC cells of human origin. One million PDX CRC cells were injected into the portal circulation of NSG mice via the spleen. Two minutes after injection, the spleen was removed using

electrocautery. When the mouse was deemed ill by increased abdominal girth, slow movement, and pale footpads, it was euthanized and the tumors were removed and sectioned in a manner similar to the subcutaneous implants. For a subset of mice (CLR4, CLR27, CLR28, CLR32) the CRC liver metastatic tumor cells were injected into the spleens of another set of NSG mice in order to obtain metastatic derivatives with enhanced ability to colonize the liver.

## Flow cytometric cell sorting and RNA sequencing

To ensure minimal contamination from mouse stromal or blood cells during RNA sequencing, we performed flow cytometric cell sorting of the PDX cell suspension after it had been processed through the magnetic-based mouse cell depletion kit (130-104-694, Miltenyi). Single cells that bound an APC-conjugated anti-human CD326 antibody (324208, BioLegend) and did not bind to a FITC-conjugated anti-mouse H-2Kd antibody (116606, BioLegend) were positively selected and considered to be PDX CRC cells. RNA was isolated from these double-sorted CRC PDX cells (37500, Norgen), ribosomal RNA was removed (MRZH11124, illumina), and the samples were prepared for RNA-sequencing using script-seq V2 (SSV21124, illumina). RNA sequencing was performed by the RU Genomics Resource Center on an Illumina HiSeq 2000 with 50 basepair single read sequencing. The sequencing data was cleaned of low quality base pairs and trimmed of linker sequences using cutadapt (v1.2) and aligned to the reference transcriptome (Hg19) using TopHat (v2). Cufflinks (v2) was used to estimate transcript abundances. Upon merging assemblies (Cuffmerge), comparison of samples was made using Cuffdiff (v2) to determine genes that were differentially expressed between parental and liver-metastatic derivative xenografts. Fisher's method was used to determine genes that were differentially expressed across all analyzed gene sets.

## Gene expression profile clustering

Correlation matrix of gene expression profiles from RNA sequencing were generated using Spearman's correlation coefficient. Clustering was performed in R using Euclidean distance and complete agglomeration method.

## Gene Set Enrichment Analysis (GSEA)

Each isogenic tumor pair (parental and liver-metastatic derivative) was evaluated for changes in the Hallmark gene sets using GSEA (v2.2.1, Broad Institute). Additionally, a composite gene set using Fisher's method as described in the section above was analyzed using GSEA.

## U-$^{13}$C-Glutamine Stable Isotope Tracing

shCTRL or shPCK1 LS174T cells were plated at $3 \times 10^5$ cells per well in a 6-well plate and allowed to adhere to the plate for 24 hr. Cells were then pre-treated with RPMI-1640 media containing 6 mM glucose at 0.5% $O_2$ for 6 hr, then replaced with RPMI-1640 media containing 2 mM U-$^{13}$C-glutamine (Cambridge Isotope Laboratories, #CLM-1822-H) for 0–6 hr at 0.5% $O_2$. Metabolites were extracted at 0, 2, 4, and 6 hr time points.

## Metabolite extraction

Metabolite extraction and subsequent Liquid-Chromatography coupled to High-Resolution Mass Spectrometry (LC-HRMS) for polar metabolites of cells was carried out using a Q Exactive Plus. shCTRL or shPCK1 LS174T were plated at $3 \times 10^5$ cells/well in triplicate with RPMI1640 + dialyzed FBS + 6 mM glucose and remained in 0.5% $O_2$ or normoxia for 24 hr. For PDX metabolite profiling, 100 mg of frozen PDXs were used. For all metabolite profiling, cells were washed with ice cold 0.9% NaCl and harvested in ice cold 80:20 LC-MS methanol:water (*v/v*). Samples were vortexed vigorously and centrifuged at 20,000 *g* at maximum speed at 4°C for 10 min. Supernatant was transferred to new tubes. Samples were then dried to completion using a nitrogen dryer. All samples were reconstituted in 30 μl 2:1:1 LC-MS water:methanol:acetonitrile. The injection volume for polar metabolite analysis was 5 μl.

## Liquid chromatography

A ZIC-pHILIC 150 × 2.1 mm (5 μm particle size) column (EMD Millipore) was employed on a Vanquish Horizon UHPLC system for compound separation at 40°C. The autosampler tray was held at 4°

C. Mobile phase A is water with 20 mM Ammonium Carbonate, 0.1% Ammonium Hydroxide, pH 9.3, and mobile phase B is 100% Acetonitrile. The gradient is linear as follows: 0 min, 90% B; 22 min, 40% B; 24 min, 40% B; 24.1 min, 90% B; 30 min, 90% B. The follow rate was 0.15 ml/min. All solvents are LC-MS grade and purchased from Fisher Scientific.

### Mass spectrometry

The Q Exactive Plus MS (Thermo Scientific) is equipped with a heated electrospray ionization probe (HESI) and the relevant parameters are as listed: heated capillary, 250°C; HESI probe, 350°C; sheath gas, 40; auxiliary gas, 15; sweep gas, 0; spray voltage, 3.0 kV. A full scan range from 55 to 825 ($m/z$) was used. The resolution was set at 70,000. The maximum injection time was 80 ms. Automated gain control (AGC) was targeted at $1 \times 10^6$ ions. Maximum injection time was 20 msec.

### Peak extraction and data analysis

Raw data collected from LC-Q Exactive Plus MS was processed on Skyline (https://skyline.ms/project/home/software/Skyline/begin.view) using a five ppm mass tolerance and an input file of $m/z$ and detected retention time of metabolites from an in-house library of chemical standards. The output file including detected $m/z$ and relative intensities in different samples was obtained after data processing. Quantitation and statistics were calculated using Microsoft Excel, GraphPad Prism 8.1, and Rstudio 1.0.143.

### Statistics

Kaplan-Meier analysis was used to evaluate patient survival based upon PDX parameters. Sample size in mouse experiments was chosen based on the biological variability observed with a given genotype. Non-parametric tests were used when normality could not be assumed. Mann Whitney test and Student's $t$ test were used when comparing independent shRNAs to shControl. One-tailed tests were used when a difference was predicted to be in one direction; otherwise, a two-tailed test was used. A $P$ value less than or equal to 0.05 was considered significant (*:p<0.05, **:p<0.01, ***:p<0.001, and ****: p<0.0001). Error bars represent SEM unless otherwise indicated.

## Acknowledgements

We thank all members of our laboratory for helpful discussions. We thank Kivanc Birsoy for expert guidance on metabolite profiling and comments on the revised manuscript. We thank Lisa Fish and Claudio Alarcon for detailed readings of the manuscript drafts. We thank Hoang Nguyen, Doowon Huh and Lisa Noble for technical assistance with in vivo experiments. We thank Gadi Lalazar and Sanford Simon for assistance with histologic photomicrographs. We thank Greg Carbonetti, Xiao-dong Huang, and Huiyong Zhao for establishing initial PDXs at the MSKCC Antitumor Assessment core facility. We thank Marissa Mattar for curation of the clinical database of patients that provided tumor samples for our study. We thank the Flow Cytometry Resource Center (Svetlana Mazel, Stanka Semova, and Selamawit Tadesse) for guidance and execution of cell sorting and the Genomics Resource Center (Connie Zhou, director) for high-throughput sequencing. We thank the Comparative Bioscience Center for quality animal husbandry.

## Additional information

### Funding

| Funder | Grant reference number | Author |
| --- | --- | --- |
| National Center for Advancing Translational Sciences | UL1 TR001866 | Norihiro Yamaguchi Ethan M Weinberg |
| Meyer Foundation | | Norihiro Yamaguchi |
| Helmsley Charitable Trust | | Norihiro Yamaguchi |
| National Institute of General Medical Sciences | T32GM07739 | Alexander Nguyen |

| NIH Office of the Director | T32CA009673-36A1 | Hani Goodarzi |
| NIH Office of the Director | 1K99CA194077-01 | Hani Goodarzi |
| National Cancer Institute | K00CA222986 | Maria V Liberti |
| Starr Foundation | | Sohail F Tavazoie |
| Black Family | Center for Research on Human Cancer Metastasis | Sohail F Tavazoie |
| HHMI | Faculty Scholar Program | Sohail F Tavazoie |

The funders had no role in study design, data collection and interpretation, or the decision to submit the work for publication.

### Author contributions

Norihiro Yamaguchi, Ethan M Weinberg, Alexander Nguyen, Conceptualization, Data curation, Investigation; Maria V Liberti, Investigation, Visualization; Hani Goodarzi, Conceptualization, Data curation, Formal analysis, Visualization; Yelena Y Janjigian, Philip B Paty, Leonard B Saltz, T Peter Kingham, Resources, Supervision; Jia Min Loo, Data curation, Writing - review and editing; Elisa de Stanchina, Resources, Methodology, Writing - review and editing; Sohail F Tavazoie, Conceptualization, Supervision

### Author ORCIDs

Norihiro Yamaguchi https://orcid.org/0000-0001-8023-9748
Alexander Nguyen https://orcid.org/0000-0001-6578-3454
Sohail F Tavazoie https://orcid.org/0000-0002-4966-9018

### Ethics

Human subjects: Approval for the study was obtained through the MSKCC Institutional Review Board/Privacy Board (protocol 10-018A), the MSKCC Institutional Animal Care and Use Committee (protocol 04-03-009), The Rockefeller University Institutional Review Board (protocol STA-0681). Written consent was obtained from all human participants who provided samples for patient-derived xenografts.

Animal experimentation: This study was performed in strict accordance with the recommendations in the Guide for the Care and Use of Laboratory Animals of the National Institutes of Health. All of the animals were handled according to approved protocol, The Rockefeller University Institutional Animal Care and Use Committee (protocol 15783-H).

### Decision letter and Author response

Decision letter https://doi.org/10.7554/eLife.52135.sa1
Author response https://doi.org/10.7554/eLife.52135.sa2

## Additional files

### Supplementary files

- Transparent reporting form

### Data availability

Sequencing data have been deposited in GEO under accession codes GSE138248.

The following dataset was generated:

| Author(s) | Year | Dataset title | Dataset URL | Database and Identifier |
|---|---|---|---|---|
| Yamaguchi N, Weinberg E | 2019 | mRNA sequencing of highly and lowly metastatic human colorectal cancer PDXs | https://www.ncbi.nlm.nih.gov/geo/query/acc.cgi?acc=GSE138248 | NCBI Gene Expression Omnibus, GSE138248 |

The following previously published datasets were used:

| Author(s) | Year | Dataset title | Dataset URL | Database and Identifier |
|---|---|---|---|---|
| Kim J, Kim S, Kim J | 2014 | Gene expression profiling study by RNA-seq in colorectal cancer | http://www.ncbi.nlm.nih.gov/geo/query/acc.cgi?acc=GSE50760 | NCBI Gene Expression Omnibus, GSE50760 |
| Ki DH, Jeung HC, Park CH, Kang SH, Lee G, Kim N, Jeung H, Rha S | 2007 | Whole genome analysis for liver metastasis gene signitures in colorectal cancer | http://www.ncbi.nlm.nih.gov/geo/query/acc.cgi?acc=GSE6988 | NCBI Gene Expression Omnibus, GSE6988 |
| Stange DE, Engel F, Radlwimmer BF, Lichter P | 2009 | Expression Profile of Primary Colorectal Cancers and associated Liver Metastases | http://www.ncbi.nlm.nih.gov/geo/query/acc.cgi?acc=GSE14297 | NCBI Gene Expression Omnibus, GSE14297 |
| Sheffer M, Bacolod MD, Zuk O, Giardina SF, Pincas H, Barany F, Paty PB, Gerald WL, Notterman DA, Domany E | 2009 | Expression data from colorectal cancer patients | http://www.ncbi.nlm.nih.gov/geo/query/acc.cgi?acc=GSE41258 | NCBI Gene Expression Omnibus, GSE41258 |

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
