## [Decision Letter]

**Acceptance summary:**

The generation of patient derived colon cancer xenograft models and their subsequent characterization, described in your paper, have provided insights into some of the mechanisms involved in metastatic colonization in patients. In particular, the finding that *PCK1* affects de novo pyrimidine nucleotide synthesis, which in turn is required for metastatic growth of tumors provides a potential pathway that could be targeted as therapeutically.

**Decision letter after peer review:**

Thank you for submitting your article "PCK1 and DHODH drive colorectal cancer liver metastatic colonization and hypoxic pyrimidine nucleotide biosynthesis" for consideration by *eLife*. Your article has been reviewed by two peer reviewers, and the evaluation has been overseen by a Reviewing Editor and Richard White as the Senior Editor. The reviewers have opted to remain anonymous.

The reviewers have discussed the reviews with one another and the Reviewing Editor has drafted this decision to help you prepare a revised submission.

Summary:

Your manuscript described the generation and characterization of patient derived colon cancer xenograft models whose metastatic potential correlates with the observed metastatic behavior of the tumors from which they were isolated. Using RNA sequencing to analyze highly metastatic variants, you found that the metabolic enzyme *PCK1* was unregulated. Suppression of *PCK1* limits in vitro cell growth and reduces nucleotide levels specifically in hypoxia. Uridine supplementation restored the viability in hypoxia in cells where *PCK1* is inhibited by RNAi. Overall this work is of potential interest.

Essential revisions:

The reviewers noted that the description of patient-derived xenografts and the identification of *PCK1* as an enzyme that contributes to tumorigenesis has already been reported. However, they felt that the role of this enzyme in metastasis merits consideration. However, it would be essential for you to address the following questions:

1) Given the prior work on *PCK1*, both reviewers request further mechanistic insights into how *PCK1* contributes to metastasis beyond what is presented in the manuscript. Is there evidence that catapleuresis is involved?

2) It is unclear what metastasizing cells and hypoxic cells have in common that make them both sensitive to pathway inhibition, while primary tumors (which may also experience hypoxia?) are not.

3) The RNAi experiments with *PCK1* are interesting but require rescue experiments to confirm that they are on target, particularly since different RNAi reagents are used in different experimental models.

4) In addition, does purine supplementation rescue the phenotype?

---

## [Author Response]

Essential revisions:The reviewers noted that the description of patient-derived xenografts and the identification of PCK1 as an enzyme that contributes to tumorigenesis has already been reported. However, they felt that the role of this enzyme in metastasis merits consideration. However, it would be essential for you to address the following questions:1) Given the prior work on PCK1, both reviewers request further mechanistic insights into how PCK1 contributes to metastasis beyond what is presented in the manuscript. Is there evidence that catapleuresis is involved?

We appreciate the reviewer’s comments and have conducted a set of experiments that have expanded the mechanistic scope of the work.

1) To further define the mechanism by which *PCK1* promotes colorectal cancer (CRC) metastasis, we conducted U-^13^C-glutamine metabolic flux analyses under hypoxia. We provide molecular evidence that *PCK1* promotes reductive carboxylation to produce aspartate and orotate in CRC cells under hypoxia in a cataplerotic manner (Figure 6G-H, Figure 6—figure supplement 1D-E).

2) We performed aspartate and pyruvate rescue experiments to functionally validate whether these metabolites, which are involved in nucleotide synthesis, are necessary for restoration of hypoxic cell growth in *PCK1* silenced cells. These experiments were performed since it has been previously shown that aspartate and pyruvate become essential for cell growth under hypoxia (Garcia-Bermudez et al., 2018). We observed that aspartate supplementation partially and significantly rescued hypoxic cell growth in *PCK1*-depleted cells. Pyruvate supplementation had a modest effect in rescuing hypoxic cell growth in *PCK1*-depleted cells. Combined supplementation with aspartate and pyruvate nearly fully rescued the hypoxic cell growth in *PCK1*-depleted cells. These findings reveal that a major role for *PCK1* is generation of aspartate as a precursor to nucleotides. Pyruvate can act as a precursor for aspartate and can replete NAD under hypoxia. The finding that combined aspartate and pyruvate provides the maximal rescue suggests that intracellular aspartate depletion is a key defect that *PCK1* over-expression overcomes under hypoxia (Figure 6—figure supplement 1F). *PCK1*-mediated overcoming of this defects enables enhanced nucleotide generation under hypoxia.

2) It is unclear what metastasizing cells and hypoxic cells have in common that make them both sensitive to pathway inhibition, while primary tumors (which may also experience hypoxia?) are not.

We thank the reviewers for these important points. Colorectal cancer cells metastasize to the liver via the portal circulation, which is hypoxemic. Moreover, the liver microenvironment itself is known to contain hypoxic regions, with metabolically active hepatocytes at the periportal region displaying high rates of oxygen consumption and hepatocytes at the perivenous region actively undergoing glycolysis (Dupuy et al., 2015; Jungermann and Kietzmann, 2000). Upon arriving in the hepatic microenvironment, CRC cells experience acute hypoxia and are poorly adapted to the liver microenvironment prior to induction of a HIF-activated response. Indeed, our prior imaging studies revealed substantial cell death upon CRC entry into the hepatic microenvironment (Loo et al., 2015).Thus, this is a major bottle-neck that leads to substantial attrition. At the primary tumor site, cells are provided oxygenated arterial blood, rather than hypoxemic venous blood. Furthermore, cancer cells at the primary site are not competing with perivenous hepatocytes for oxygen consumption. We have better referenced the past literature describing the physiological basis for the hypoxic nature of the hepatic microenvironment.

3) The RNAi experiments with PCK1 are interesting but require rescue experiments to confirm that they are on target, particularly since different RNAi reagents are used in different experimental models.

In the revised manuscript, we have now included data showing that expressing *PCK1* in CRC cells expressing an shRNA targeting the 3’UTR of *PCK1* fully rescued hypoxic cell growth. This validates the specificity of our observations with multiple distinct RNAi’s as well as pharmacology (Figure 6—figure supplement 1A).

4) In addition, does purine supplementation rescue the phenotype?

We appreciate the important comments by the reviewers. Montal et al. had shown that *PCK1* promotes CRC cell growth by increasing ribose abundance, which they proposed was the mechanism underlying enhanced purine synthesis (Montal et al., 2015). To test the importance of pyrimidines and purines/ribose in hypoxic cancer cell growth, we conducted hypoxic cell growth assays with purines, ribose, and uridine supplementation (Figure 6I). While purines, ribose, or the combination of purines and ribose had modest effects on hypoxic cell growth, uridine, a pyrimidine nucleoside, fully rescued hypoxic cell growth in *PCK1* silenced cells suggesting that *PCK1* is primarily overcoming the barrier of pyrimidine nucleotide depletion under hypoxia.